# Expansion of intestinal *Prevotella copri* correlates with enhanced susceptibility to arthritis

Jose U Scher[1†], Andrew Sczesnak[2,3†], Randy S Longman[2,4†], Nicola Segata[5,6], Carles Ubeda[7,8], Craig Bielski[6], Tim Rostron[9], Vincenzo Cerundolo[9], Eric G Pamer[7], Steven B Abramson[1], Curtis Huttenhower[6], Dan R Littman[2,10]*

[1]Department of Medicine, New York University School of Medicine and Hospital for Joint Diseases, New York, United States; [2]Molecular Pathogenesis Program, The Kimmel Center for Biology and Medicine of the Skirball Institute, New York University School of Medicine, New York, United States; [3]Graduate Program in Bioinformatics and Computational Biology, University of California, San Francisco, San Francisco, United States; [4]Jill Roberts IBD Center, Department of Medicine, Weill Cornell Medical College, New York, United States; [5]Centre for Integrative Biology, University of Trento, Trento, Italy; [6]Department of Biostatistics, Harvard School of Public Health, Boston, United States; [7]Immunology Program, Infectious Diseases Service, and The Lucille Castori Center for Microbes, Inflammation, and Cancer, Memorial Sloan-Kettering Cancer Center, New York, United States; [8]Centro Superior de Investigacion en Salud Publica, University of Valencia, Valencia, Spain; [9]Department of Medicine, Weatherall Institute of Molecular Medicine, University of Oxford, Oxford, United Kingdom; [10]Howard Hughes Medical Institute, New York University School of Medicine, New York, United States

*For correspondence: dan. littman@med.nyu.edu

†These authors contributed equally to this work

Competing interests: The authors declare that no competing interests exist.

**Abstract** Rheumatoid arthritis (RA) is a prevalent systemic autoimmune disease, caused by a combination of genetic and environmental factors. Animal models suggest a role for intestinal bacteria in supporting the systemic immune response required for joint inflammation. Here we performed 16S sequencing on 114 stool samples from rheumatoid arthritis patients and controls, and shotgun sequencing on a subset of 44 such samples. We identified the presence of *Prevotella copri* as strongly correlated with disease in new-onset untreated rheumatoid arthritis (NORA) patients. Increases in *Prevotella* abundance correlated with a reduction in *Bacteroides* and a loss of reportedly beneficial microbes in NORA subjects. We also identified unique *Prevotella* genes that correlated with disease. Further, colonization of mice revealed the ability of *P. copri* to dominate the intestinal microbiota and resulted in an increased sensitivity to chemically induced colitis. This work identifies a potential role for *P. copri* in the pathogenesis of RA.

## Introduction

Rheumatoid arthritis (RA) is a highly prevalent systemic autoimmune disease with predilection for the joints. If left untreated, RA can lead to chronic joint deformity, disability, and increased mortality. Despite recent advances towards understanding its pathogenesis (*Mcinnes and Schett, 2011*), the etiology of RA remains elusive. Many genetic susceptibility risk alleles have been discovered and validated (*Stahl et al., 2010*) but are insufficient to explain disease incidence. RA is therefore a complex (multifactorial) disease requiring both environmental and genetic factors for onset (*Mcinnes and Schett, 2011*).

**eLife digest** We share our bodies with a diverse set of microorganisms, known collectively as the human microbiome. Indeed, estimates suggest that our bodies contain 10 times as many microbial cells as human cells. Our stomach and intestines alone are home to many hundreds and possibly thousands of microbial species that break down indigestible compounds and help to prevent the growth of harmful bacteria. The immune system must therefore learn to tolerate these microorganisms, while retaining the ability to launch attacks against microorganisms that cause harm. Failure of this process may increase the risk of autoimmune diseases in which the body mistakenly attacks its own cells and tissues.

Rheumatoid arthritis is a chronic autoimmune disease marked by inflammation of the joints. Although the causes of rheumatoid arthritis are unknown, mice with mutations that increase the risk of the disease remain healthy if they are kept under sterile conditions. However, if these mice are exposed to certain species of bacteria sometimes found in the gut, they begin to show signs of joint inflammation.

Here, Scher et al. used genome sequencing to compare gut bacteria from patients with rheumatoid arthritis and healthy controls. A bacterial species called *Prevotella copri* was more abundant in patients suffering from untreated rheumatoid arthritis than in healthy individuals. Moreover, the presence of *P. copri* corresponded to a reduction in the abundance of other bacterial groups—including a number of beneficial microbes. In a mouse model of gut inflammation, animals colonized with *P. copri* had more severe disease than controls, consistent with a pro-inflammatory function of this organism.

Current treatments for rheumatoid arthritis target symptoms. However, by highlighting the role played by gut bacteria, the work of Scher et al. suggests that novel treatment options focused on curbing the spread of *P. copri* in the gut could delay or prevent the onset of this disease.

Among environmental factors, the intestinal microbiota has emerged as a possible candidate responsible for the priming of aberrant systemic immunity in RA (*Scher and Abramson, 2011*). The microbiota encompasses hundreds of bacterial species whose products represent an enormous antigenic burden that must largely be compartmentalized to prevent immune system activation (*Littman and Pamer, 2011*). In the healthy state, intestinal lamina propria cells of both innate and adaptive immune systems cooperate to maintain physiological homeostasis. In RA, there is increased production of both self-reactive antibodies and pro-inflammatory T lymphocytes. Although mechanisms for targeting of synovium by inflammatory cells have not been fully elucidated, studies in animal models suggest that both T cell and antibody responses are involved in arthritogenesis. Moreover, an imbalance in the composition of the gut microbiota can alter local T-cell responses and modulate systemic inflammation. Mice rendered deficient for the microbiota (germ-free) lack pro-inflammatory Th17 cells, and colonization of the gastrointestinal tract with segmented filamentous bacteria (SFB), a commensal microbe commonly found in mammals, is sufficient to induce accumulation of Th17 cells in the lamina propria (*Ivanov et al., 2009*; *Sczesnak et al., 2011*).

In several animal models of arthritis, mice are persistently healthy when raised in germ-free conditions. However, the introduction of specific gut bacterial species is sufficient to induce joint inflammation (*Rath et al., 1996*; *Abdollahi-Roodsaz et al., 2008*; *Wu et al., 2010*), and antibiotic treatment both prevents and abrogates a rheumatoid arthritis-like phenotype in several mouse models. Upon mono-colonization of arthritis-prone K/BxN mice with SFB, the induced Th17 cells potentiate inflammatory disease (*Wu et al., 2010*). An imbalance in intestinal microbial ecology, in which SFB is dominant, may result in reduced proportions or functions of anti-inflammatory regulatory T cells (Treg) and a predisposition towards autoimmunity. This appears to affect not only the local immune response, but also systemic inflammatory processes, and may explain, at least in part, reduced Treg cell function in RA patients (*Zanin-Zhorov et al., 2010*). Thus, T cells whose functions are dictated by intestinal commensal bacteria can be effectors of pathogenesis in tissue-specific autoimmune disease.

Although recent studies of the human microbiome (*Arumugam et al., 2011*; *Human Microbiome Project Consortium, 2012*) have characterized the composition and diversity of the healthy gut microbiome, and disease-associated studies revealed correlations between taxonomic abundance and some clinical phenotypes (*Frank et al., 2011*; *Morgan et al., 2012*; *Qin et al., 2012*), a role for distinct microbial

taxa and metagenomic markers in systemic inflammatory disease has not been defined. While treatment with antibiotics has been a therapeutic modality in RA for decades, no microbial organism has been shown to be associated with the disease. Based on the discovery that SFB-induced Th17 cells directly contribute to the onset of arthritis in gnotobiotic mice (*Wu et al., 2010*), we analyzed the fecal microbiota in patients with RA. We used 16S ribosomal RNA gene sequencing to classify the microbiota in patients with new-onset (untreated) RA, chronic (treated) RA, psoriatic arthritis, and age- and ethnicity-matched healthy controls. We found a marked association of *Prevotella copri* with new-onset RA (NORA) patients and not with other patient groups. Shotgun sequencing of the microbiome indicated that some *P. copri* genes are differentially present in NORA-associated and healthy samples. Colonization of mice with *P. copri* enhanced susceptibility to chemical colitis, consistent with a pro-inflammatory potential of this organism. Taken together, our results suggest that NORA-associated *P. copri* may contribute to the pathogenesis of human arthritis.

## Results

### Association of *Prevotella* with new-onset rheumatoid arthritis

To determine if particular bacterial clades are associated with rheumatoid arthritis, we performed sequencing of the 16S gene (regions V1–V2, 454 platform) on 114 fecal DNA samples—44 samples collected from NORA patients at time of initial diagnosis and prior to immunosuppressive treatment, 26 samples from patients with chronic, treated rheumatoid arthritis (CRA), 16 samples from patients with psoriatic arthritis (PsA), and 28 samples from healthy controls (HLT) (*Table 1*). Sequences were analyzed with MOTHUR (*Schloss et al., 2009*) to cluster operational taxonomic units (OTUs, species level classification) at a 97% identity threshold, assign taxonomic identifiers, and calculate clade relative abundances. Although PsA patients revealed a reduction in sample diversity similar to that of IBD

**Table 1.** Demographic and clinical data among subjects with new-onset rheumatoid arthritis (NORA), chronic, treated rheumatoid arthritis (CRA), psoriatic arthritis (PsA), and healthy controls (HLT)

| | NORA (n = 44) | CRA (n = 26) | PsA (n = 16) | Healthy (n = 28) |
|---|---|---|---|---|
| Age, years, mean (median) | 42.4 (40.0) | 50.0 (49.0) | 46.3 (46.0) | 42.8 (40.0) |
| Female, % | 75 | 88 | 56 | 75 |
| Disease duration, months, mean (median) | 5.4 (2.0) | 72.3 (48.0) | 0.8 (0.0) | N/A |
| Disease activity parameters | | | | |
| ESR, mm/h, mean | 34.6 | 33.5 | 19.7 | 10.2 |
| CRP, mg/l, mean | 20.6 | 8.2 | 7.6 | 1.1 |
| DAS28, mean (median) | 5.4 (5.7) | 4.7 (5.0) | 4.8 (4.7) | N/A |
| Patient VAS pain, mm, mean (median) | 61.4 (57.5) | 51.5 (62.5) | 50.6 (45.0) | N/A |
| TJC-28, mean (median) | 11.2 (8.5) | 7.6 (7.0) | 8.8 (6.5) | N/A |
| SJC-28, mean (median) | 8.3 (8.0) | 4.6 (3.0) | 4.8 (3.0) | N/A |
| Autoantibody status | | | | |
| IgM-RF positive, % | 95 | 81 | 13 | 11 |
| ACPA positive, % | 100 | 85 | 6 | 7 |
| IgM-RF and/or ACPA positive, % | 100 | 96 | 13 | 14 |
| IgM-RF titer, kU/l, mean (median) | 341.3(157.0) | 178.2 (89.0) | 3.6 (0.0) | 20.5 (0.0) |
| ACPA titer, kAU/l, mean (median) | 117.6 (114.0) | 90.8 (57.0) | 1.6 (0.0) | 9.6 (0.0) |
| Medication use | | | | |
| Methotrexate, % | 0 | 42 | 6 | 0 |
| Prednisone, % | 0 | 15 | 6 | 0 |
| Biological agent, % | 0 | 12 | 0 | 0 |

patients (*Morgan et al., 2012*), diversity was comparable between NORA, CRA and healthy groups at 3.02 +/− 0.66 (mean, SD) overall by Shannon Diversity Index (*Figure 1—figure supplement 1A*). However, when applying Simpson's Dominance Index, the NORA group was less diverse (*Figure 1—figure supplement 1B*), suggesting that these patients harbored a relatively higher abundance of common taxa. Analysis at the major taxonomic hierarchy levels showed no significant differences in either phyla abundance or the ratio of Bacteroidetes/Firmicutes (*Figure 1—figure supplement 1C*) between all groups. At the level of family abundances, however, we noted a significant enrichment of Prevotellaceae in NORA subjects (*Figure 1A*, *Figure 1—figure supplement 1D*). Using the linear

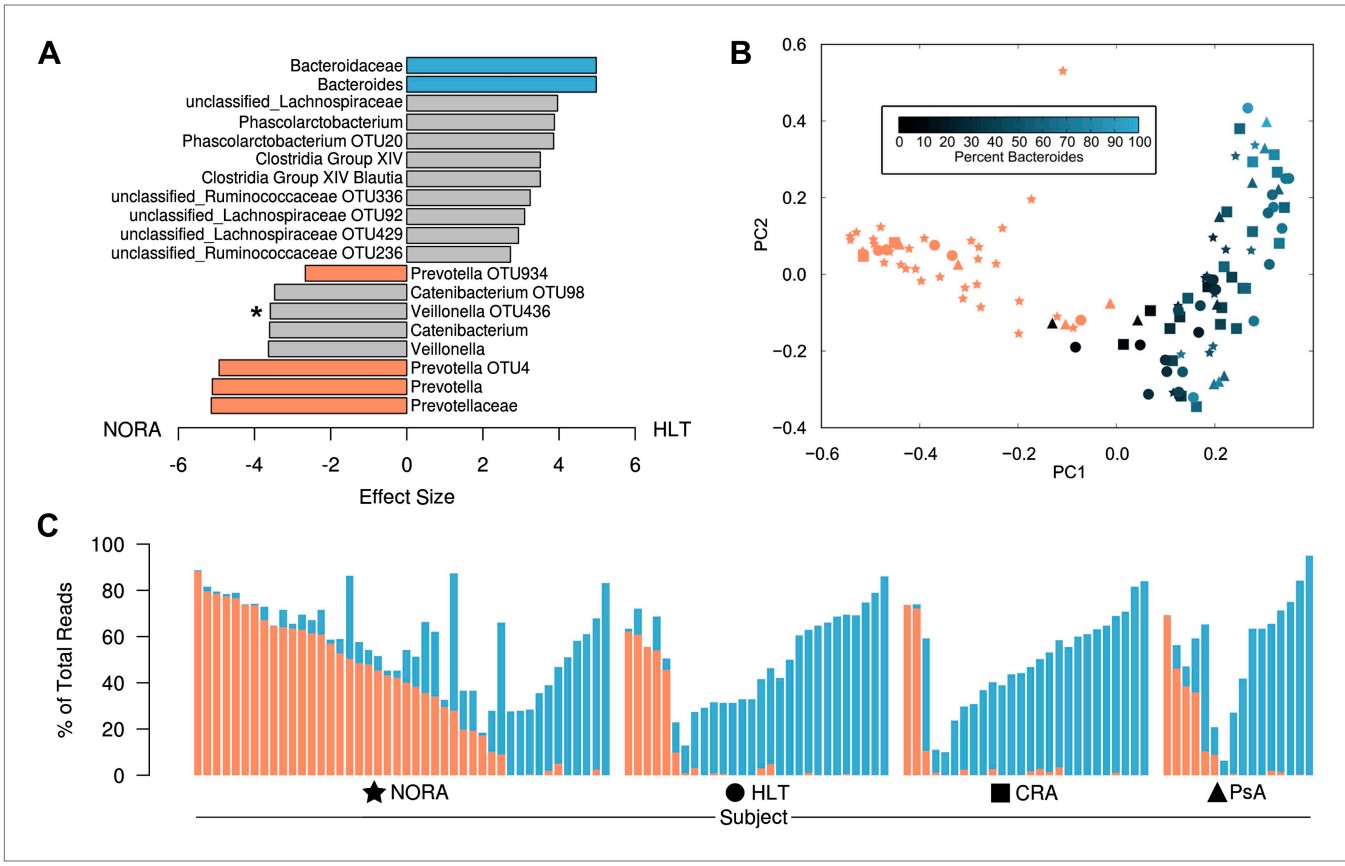

**Figure 1**. Differences in the relative abundance of *Prevotella* and *Bacteroides* in 114 subjects with and without arthritis, determined by 16S sequencing (regions V1–V2, 454 platform). (**A**) LEfSe (*Segata et al., 2011*) was used to compare the abundances of all detected clades among all groups, producing an effect size for each comparison ('Materials and methods'). All results shown are highly significant (q<0.01) by Kruskal-Wallis test adjusted with the Benjamini-Hochberg procedure for multiple testing, except that indicated with an asterisk, which is significant at q<0.05. Negative values (left) correspond to effect sizes representative of NORA groups, while positive values (right) correspond to effect sizes in HLT subjects. *Prevotella* was found to be over-represented in NORA patients, while *Bacteroides* was over-represented in all other groups. (**B**) The Bray-Curtis distance between all subjects was calculated and used to generate a principal coordinates plot in MOTHUR (*Schloss et al., 2009*). The first two components are shown. Subjects with an abundance of *Prevotella* greater than 10% were colored red. Other subjects were colored according to their *Bacteroides* abundance as shown. NORA subjects (stars) primarily cluster together according to their *Prevotella* abundance, and the x-axis is representative of differences in the relative abundance of *Prevotella* and *Bacteroides*. (**C**) The abundances of *Prevotella* (red) and *Bacteroides* (blue) are shown for all subjects, sorted in order of decreasing *Prevotella* abundance (>5%) and increasing *Bacteroides* abundance.

The following source data and figure supplements are available for figure 1:

**Source data 1**. Intermediate data and analysis tools for *Figure 1*.

**Source data 2**. Intermediate data and analysis tools for *Figure 1—figure supplement 1*.

**Figure supplement 1**. Gut microbiota richness, diversity and relative abundance in NORA patients and controls.

discriminant effect size method (LEfSe, see 'Materials and methods') (*Segata et al., 2011*) to compare detected clades (33 families, 177 genera, 996 OTUs) among all groups, we found a positive association of two specific *Prevotella* OTUs with NORA and an inverse correlation with Group XIV Clostridia, Lachnospiraceae, and *Bacteroides* as compared to healthy controls (*Figure 1A*). Of all detected Prevotellaceae OTUs, OTU4 was the most highly represented with 171,486 supporting reads at 11.49 +/− 17.85 (mean, SD) percent of reads per sample. OTU12, the next most abundant Prevotellaceae, was supported by 12,119 reads at 2.00 +/− 5.42 (mean, SD) percent of reads per sample. Other Prevotellaceae OTUs (including *Prevotella* OTU934) were more scarcely represented with 1,232 +/− 2,305 (mean, SD) total supporting reads at less than 0.5% total reads per sample. We therefore reasoned that OTU4 was the dominant *Prevotella* in our cohort with sixfold more supporting reads than the next most abundant OTU. Principal coordinate analysis with Bray-Curtis distances demonstrated that subjects form distinct clusters, irrespective of health or disease status (*Figure 1B*). The largest component of microbial variation corresponded to the carriage (or absence) of *Prevotella*, which significantly differentiated NORA subjects from healthy controls and other forms of arthritis. Consistent with other reports of either high *Prevotella* or high *Bacteroides* relative abundance, but rarely a high relative abundance of both, (*Faust et al., 2012*; *Yatsunenko et al., 2012*), we found segregation of *Prevotella* or *Bacteroides* dominance in the intestinal microbiome (*Figure 1C*).

To taxonomically identify *Prevotella* OTU4, OTU12, and OTU934, we generated a phylogenetic tree using the consensus 16S sequences of these OTUs and matched regions from known *Prevotella* taxa (*Figure 2—figure supplement 1*). The analysis revealed these OTUs to cluster tightly with *Prevotella copri*, a microbe isolated from human feces (*Hayashi et al., 2007*) and sequenced as part of the HMP's reference genome initiative. To further characterize *Prevotella* OTU4, the most abundant taxon, we selected four high abundance NORA samples (028B, 030B, 061B, and 089B) for shotgun sequencing (single-end, 454 platform). The resulting long reads were used to generate metagenomic assemblies (*Table 2*, see 'Materials and methods') which served as input to PhyloPhlAn (*Segata et al., 2013*). Briefly, PhyloPhlAn locates 400 ubiquitous bacterial genes in a given assembly by sequence alignment in amino acid space, then builds a tree by concatenating the most discriminative positions in each gene into a single long sequence and applying FastTree (*Price et al., 2010*), a standard tree reconstruction tool. This produced a phylogenomic tree placing the taxon most represented in each sample's metagenomic contigs (i.e., *Prevotella* OTU4) again in close association with *Prevotella copri* (*Figure 2A*). We therefore chose to filter the resulting metagenomic assemblies by alignment to the *P. copri* reference genome to generate draft patient-derived genome assemblies (see 'Materials and methods'). Comparison of these draft assemblies to reference *P. copri* and to one another revealed a high degree of similarity, with possible genome rearrangements (*Figure 2B*).

Overall, 75% (33/44) of NORA patients and 21.4% (6/28) of healthy controls carried *P. copri* in their intestinal microbiota compared to 11.5% (3/26) and 37.5% (6/16) in CRA and PsA patients, respectively, at a threshold for presence of >5% relative abundance. The prevalence of *P. copri* in NORA compared to CRA, PsA, and healthy controls was statistically significant by chi-squared test, but was not significant in pairwise comparisons of the latter three cohorts (*Table 3*).

**Table 2.** Draft genome assembly statistics of four subjects with a high abundance of *Prevotella* OTU4

| Subject ID | Group | *Prevotella* OTU4 abundance (%) | # reads | Total # of contigs | Total Size (Mb) | Total N50 (kb) | Total Mean depth | *P. copri* aligned # of contigs | *P. copri* aligned Size (Mb) | *P. copri* aligned N50 (kb) | *P. copri* aligned Mean depth |
|---|---|---|---|---|---|---|---|---|---|---|---|
| 028B | NORA | 27.7 | 1,240,515 | 19,988 | 23.24 | 1.45 | 6.13 | 115 | 3.21 | 59.84 | 36.76 |
| 030B | NORA | 50.9 | 1,041,546 | 21,579 | 17.35 | 1.01 | 6.97 | 232 | 2.60 | 16.18 | 44.14 |
| 061B | NORA | 66.5 | 1,209,392 | 9,241 | 12.8 | 1.58 | 9.88 | 74 | 3.23 | 79.98 | 172.64 |
| 089B | NORA | 56.3 | 1,395,872 | 12,112 | 23.47 | 4.64 | 23.12 | 1,963 | 3.96 | 3.19 | 30.39 |
| Ref. genome | – | – | – | – | – | – | – | 83 | 3.51 | 131.4 | – |

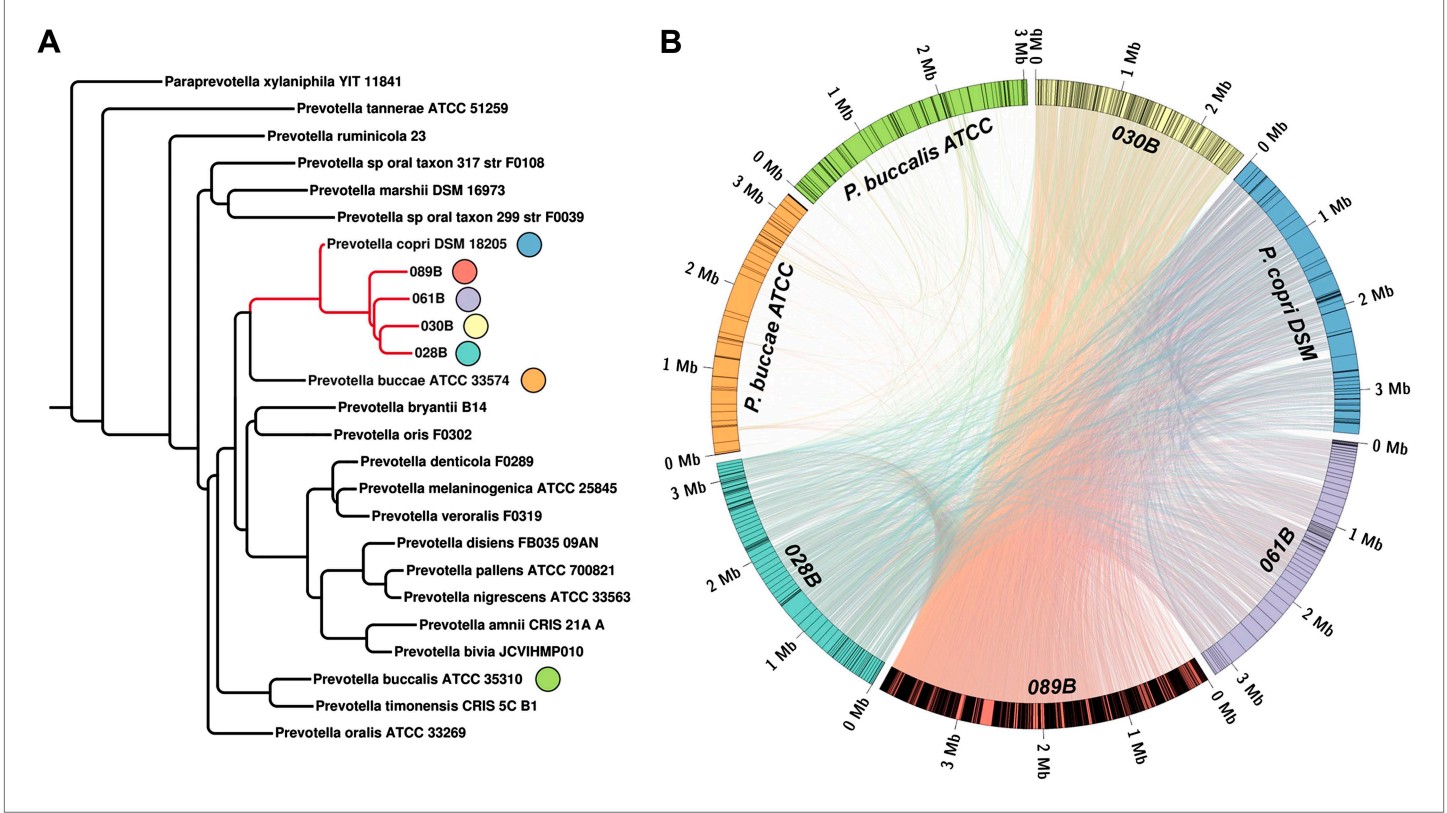

**Figure 2**. Homology-based classification of patient-associated *Prevotella*. Four NORA subjects with a high abundance of *Prevotella* OTU4 were selected for shotgun sequencing and metagenome assembly. (**A**) The resulting metagenomic contigs were used to generate a phylogenomic tree with PhyloPhlAn (*Segata et al., 2013*). (**B**) Assemblies were filtered by alignment to the reference *Prevotella copri DSM 18205* genome, keeping contigs with at least one 300 bp region aligned at 97% identity or greater. The resulting draft patient-derived *P. copri* assemblies were aligned to one another, the reference *P. copri* genome, and two distinct *Prevotella* taxa (*Prevotella buccae* and *Prevotella buccalis*). Colored arcs represent assemblies as labeled, lines connecting arcs represent regions of >97% identity >1 kb in length, and gray lines dividing colored arcs represent boundaries between contigs. These results demonstrate that *Prevotella* OTU4, OTU12, and OTU934 form a clade with *P. copri* (left, red highlighted subtree) that is genetically distinct from more distant *Prevotella* taxa.

The following source data and figure supplements are available for figure 2:

**Source data 1**. Intermediate data and analysis tools for *Figure 2*.

**Source data 2**. Intermediate data and analysis tools for *Figure 2—figure supplement 1*.

**Figure supplement 1**. The representative 16S sequenced reads for *Prevotella* OTU4, OTU12, and OTU934 were aligned with MUSCLE (*Edgar, 2004*) and clustered with FastTree (*Price et al., 2010*)

## *P. copri* strains are variable and potentially diagnostic

Although initial shotgun sequencing of the patient-derived strains showed their similarity to *P. copri*, there were notable differences observed in assembled genomes upon comparison with the *P. copri* reference genome. This observation suggested that the presence or absence of particular genes in these strains might correlate with health or disease phenotypes in this cohort. To address this question, we performed shotgun sequencing on fecal DNA from NORA and healthy subjects, and chose to compare *Prevotella* sequences from 18 NORA *Prevotella*-positive subjects, which allowed for a depth of at least 7 M *Prevotella*-aligned reads (paired-end, 100 nt, Illumina platform), to those of *P. copri* from 17 healthy subjects (including 15 from the HMP database and 2 HLT from our cohort) (*Supplementary file 1A*). Samples sequenced to a depth of less than 7 M such reads were excluded (*Figure 3—figure supplement 1C*), having insufficient depth for complete recovery of *P. copri* ORFs (see 'Materials and methods').

**Table 3.** Statistical comparisons of *Prevotella copri* prevalence between cohort groups

| Comparison | Prevalence #1 | Prevalence #2 | Chi-squared p-value | Fisher's exact p-value |
|---|---|---|---|---|
| *NORA vs HLT | 33/44 | 6/28 | 2.612e-05 | 1.025e-05 |
| *NORA vs CRA | 33/44 | 3/26 | 1.031e-06 | 2.551e-07 |
| †NORA vs PsA | 33/44 | 6/16 | 0.01698 | 0.013 |
| HLT vs CRA | 6/28 | 3/26 | 0.5425 | 0.4704 |
| HLT vs PsA | 6/28 | 6/16 | 0.4239 | 0.3032 |
| CRA vs PsA | 3/26 | 6/16 | 0.1087 | 0.06282 |

*p<0.01.
†p<0.05.

First, we examined the coverage of the *P. copri* reference genome by all subjects, as an indicator of inter-individual strain variability (*Human Microbiome Project Consortium, 2012*). Overall, coverage was similar between healthy and NORA subjects in all but a few regions (*Figure 3A*, blue and red horizontal lines). Eight regions were poorly covered in all subjects with mean coverage below the 25th percentile of 0.79 FPKM, while several regions showed substantial variability between individuals (*Figure 3A*, gray vertical lines). To determine if the presence or absence of these regions within individuals was consistent between samplings, we applied MetaPhlAn (*Segata et al., 2012*) to *Prevotella*-positive HMP samples collected over multiple visits (*Figure 3B*). Briefly, MetaPhlAn determines the presence or absence of metagenomic marker genes that are specific to particular bacterial clades by analyzing the coverage of such genes by sequenced reads. Genes are called specific for a bacterial clade if they are not found in any reference genomes outside the clade, but are found in all such genomes within the clade. In concordance with a previous report (*Schloissnig et al., 2013*) documenting the temporal stability of metagenomic SNP patterns in individuals, we found that carriage of *P. copri* genes within an individual varied little between samplings. In addition to a stable set of *P. copri* core marker genes common to all samples, a subset of variable marker genes was observed to co-occur in islands across the *P. copri* genome, suggesting genomic rearrangements as a mechanism of variability (*Figure 3A*, blue boxes below plot). Together, these results suggest that *P. copri* strains vary between individuals and retain their individuality over time.

Next, we assembled a catalog of *P. copri* genes present across many individuals (i.e., the *P. copri* pangenome), by performing de novo metagenome assembly and gene calling on a per-sample basis (see 'Materials and methods'). To determine if any ORFs were differentially present in NORA subjects as compared to healthy controls, we first reduced the set of interrogated ORFs by filtering partially assembled (i.e., containing gaps, lacking stop codons), short (i.e., less than 300 bp), and low-coverage (i.e., present in fewer than five subjects) ORFs to yield a final set of 3,291 high-confidence *P. copri* ORFs (*Figure 3—figure supplement 1*). We found two ORFs differentially present in healthy controls, and 17 ORFs differentially present in NORA (*Figure 3C*; *Supplementary file 1B*). The two healthy-specific ORFs appear on the same metagenomic contig, encoding a nearly-complete *nuo* operon for NADH:ubiquinone oxidoreductase (*Figure 3—figure supplement 2A*), adjacent to a *Bacteroides* conjugative transposon. Similarly, two of the NORA-specific ORFs appear together on another metagenomic contig, encoding an ATP-binding cassette iron transporter (*Figure 3—figure supplement 2B*). These ORFs may represent good biomarkers for discrimination between healthy and disease-associated microbiota in the population at risk for RA.

## Functional potential of the NORA metagenome

To determine if the NORA metagenome encodes unique functions compared to healthy subjects, we applied HUMAnN (*Abubucker et al., 2012*) to quantitate the coverage and abundances of KEGG (*Kanehisa and Goto, 2000*) modules (small sets of genes in well-defined metabolic pathways) in healthy controls (n = 5) and a representative set of NORA subjects (n = 14) with and without *Prevotella*. We then applied LEfSe (*Segata et al., 2011*) to find statistically significant differences between groups. This analysis revealed a low abundance of vitamin metabolism (i.e., biotin, pyroxidal, and folate) and pentose phosphate pathway modules in NORA, consistent with a lack of these functions in *Prevotella* genomes (*Figure 4*). At the coverage level (presence or absence), the NORA metagenome is defined

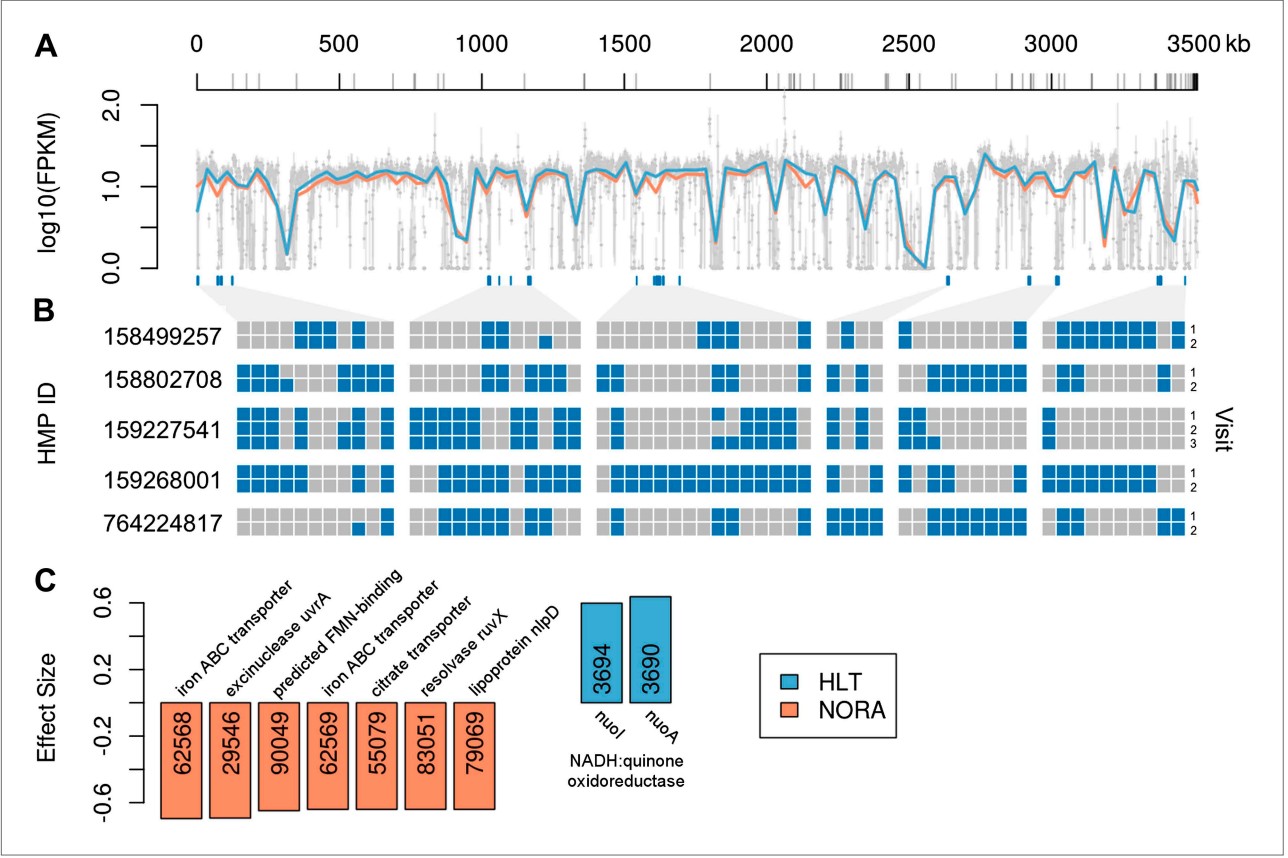

**Figure 3**. Comparison of *P. copri* genomes from healthy and NORA subjects. (**A**) Comparative coverage of the draft *P. copri DSM 18205* genome between individuals and within healthy and NORA groups. Gray points are median fragments per kilobase per million (FPKM) for 1-kb windows, gray lines within the plot are the interquartile range for each window, red and blue lines the LOWESS-smoothed average for NORA and healthy groups, respectively. Gray lines on the horizontal axis represent boundaries between assembled contigs. Regions are variably covered between subjects and groups, with several genomic islands lacking overall or especially variable (dark blue lines below the plot). (**B**) The presence (blue) or absence (gray) of previously-reported *P. copri*-unique marker genes (***Segata et al., 2012***) in 11 stool samples from five subjects of the Human Microbiome Project (HMP) are shown as a heatmap. We report, in columns, only those *P. copri*-specific markers showing variable presence/absence patterns across the considered HMP samples. Each row represents a different sample collection date, groups of rows represent subjects, and groups of columns correspond to different variably covered genomic islands. Strains of *P. copri* are defined by the presence and absence of particular genes, which remain stable for at least 6 months in these individuals. All inter- and intra-individual comparisons between rows are highly statistically significant (p<<0.001, ' Materials and methods'). (**C**) The *P. copri* pangenome was identified by finding *P. copri* ORFs in all HMP and NORA cohort subjects, and the presence or absence of these ORFs was calculated for each subject ('Materials and methods', ***Figure 3—figure supplement 1***). Several ORFs are statistically significant biomarkers between healthy and NORA status (q<0.25) (***Supplementary file 1B***, 'Materials and methods').

The following source data and figure supplements are available for figure 3:

**Source data 1**. Intermediate data and analysis tools for ***Figure 3***.

**Source data 2**. Intermediate data and analysis tools for ***Figure 3—figure supplement 1***.

**Figure supplement 1**. Recovery of *P. copri* pangenome from HMP/RA shotgun reads and determination of presence/absence of *P. copri* ORFs by alignment of reads to pangenome gene catalog.

**Figure supplement 2**. Metagenomic context of discriminative biomarker ORFs.

by an absence of functions present in *Bacteroides* and Clostridia, clades typically found in low abundance in *Prevotella*-high NORA subjects.

*Prevotella* and *Bacteroides* are closely related both functionally and phylogenetically, yet, surprisingly, are rarely found together in high relative abundance despite their ability to dominate

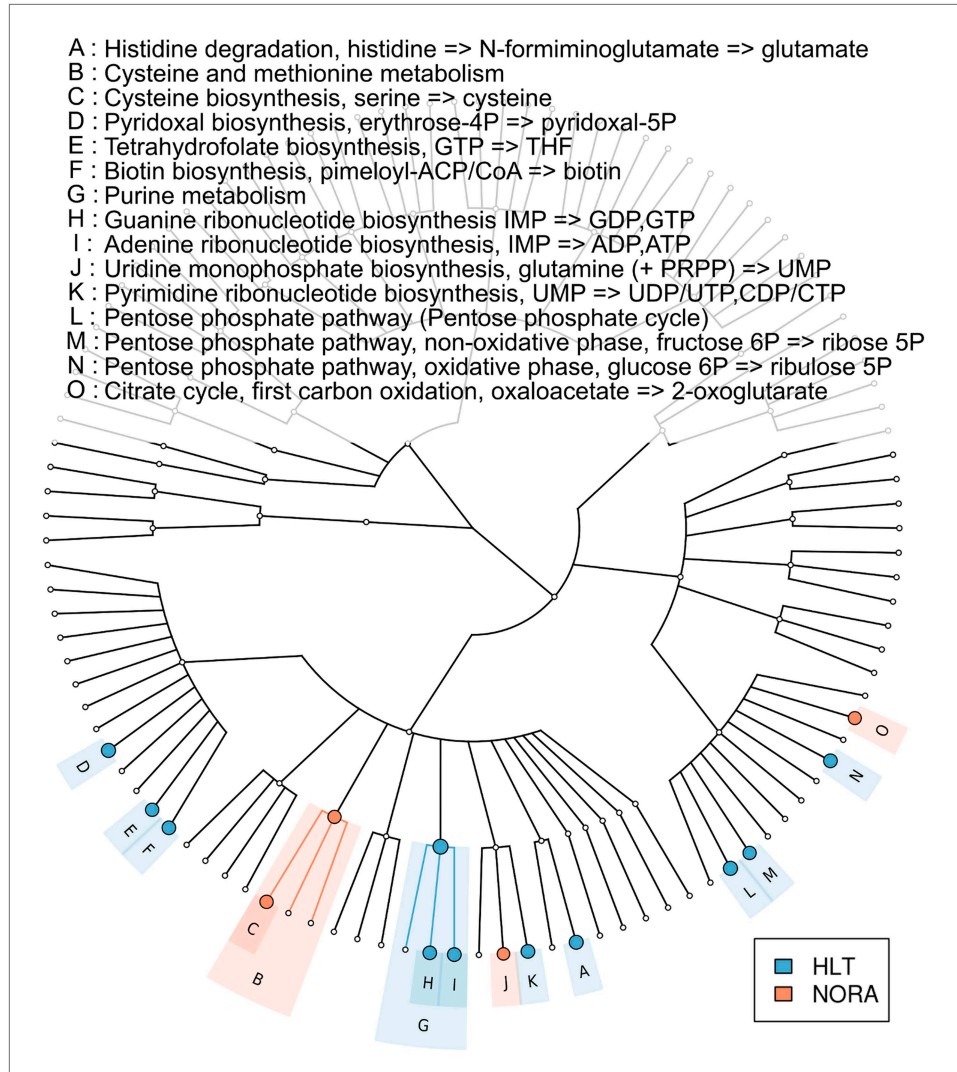

A : Histidine degradation, histidine => N-formiminoglutamate => glutamate
B : Cysteine and methionine metabolism
C : Cysteine biosynthesis, serine => cysteine
D : Pyridoxal biosynthesis, erythrose-4P => pyridoxal-5P
E : Tetrahydrofolate biosynthesis, GTP => THF
F : Biotin biosynthesis, pimeloyl-ACP/CoA => biotin
G : Purine metabolism
H : Guanine ribonucleotide biosynthesis IMP => GDP,GTP
I : Adenine ribonucleotide biosynthesis, IMP => ADP,ATP
J : Uridine monophosphate biosynthesis, glutamine (+ PRPP) => UMP
K : Pyrimidine ribonucleotide biosynthesis, UMP => UDP/UTP,CDP/CTP
L : Pentose phosphate pathway (Pentose phosphate cycle)
M : Pentose phosphate pathway, non-oxidative phase, fructose 6P => ribose 5P
N : Pentose phosphate pathway, oxidative phase, glucose 6P => ribulose 5P
O : Citrate cycle, first carbon oxidation, oxaloacetate => 2-oxoglutarate

**Figure 4**. Metabolic pathway representation in the microbiome of healthy and NORA subjects. HUMAnN (**Abubucker et al., 2012**) was applied to metagenomic reads (paired-end, 100 nt, Illumina platform) from NORA subjects (n = 14) and healthy controls (n = 5) to quantitate the abundances of hierarchically related KEGG modules in these samples ('Materials and methods' and **Supplementary file 1A**). LEfSe (**Segata et al., 2011**) was used to find statistically significant differences between groups at an alpha cutoff of 0.001 and an effect size cutoff of 2.0. Results shown here are highly significant (p<0.001) and represent large differences between groups. Modules highlighted in red are over-abundant in NORA samples while modules highlighted in blue are over-abundant in healthy samples. *Prevotella*-dominated NORA metagenomes have a dearth of genes encoding vitamin and purine metabolizing enzymes, and an excess of cysteine metabolizing enzymes.

The following source data are available for figure 4:

**Source data 1**. Intermediate data and analysis tools for **Figure 4**.

the gut microbiome individually (**Faust et al., 2012**). We hypothesized that there might be a genetic difference in these two clades that could account for their apparent co-exclusionary relationship. We therefore sought to find genes differentially present in *P. copri* but not in any of the most abundant *Bacteroides* species. This revealed K05919 (superoxide reductase), K00390 (phosphoadenosine phosphosulfate reductase), and several transporters as uniquely present in *P. copri* (**Supplementary file 1C**), and also a set of genes absent in *P. copri* but present in *Bacteroides* (**Supplementary file 1D**).

## Relative abundance of *P. copri* in NORA inversely correlates with presence of shared-epitope risk alleles

Certain alleles within the human leukocyte-antigen (HLA) Class II locus confer higher risk of disease, in particular those belonging to DRB1 (i.e., 'shared epitope' alleles or SE) (*du Montcel et al., 2005*; *Gregersen et al., 1987*). To determine whether a higher abundance of *P. copri* is associated with the host genotype, we carried out HLA sequencing on DNA from all participants in our study (*Supplementary file 1E*). Consistent with recently published mouse data (*Gomez et al., 2012*), the presence of SE alleles correlated with the composition of the gut microbiota. A subgroup analysis of NORA patients and healthy controls according to presence (or absence) of SE alleles revealed a significantly higher relative abundance of *P. copri* in those subjects lacking predisposing genes (*Figure 5*, $p<0.001$ in NORA, $p<0.05$ in HLT, 'Materials and methods').

## *P. copri* exacerbates colitis in mice

To determine if the *Prevotella*-associated metagenome is sufficient to predispose to increased inflammatory responses, antibiotic-treated C57BL/6 mice were colonized with *P. copri* by oral gavage. Analysis of DNA extracted from fecal samples 2 weeks post-gavage revealed robust colonization with *P. copri* (*Figure 6A*). Sequencing of the 16S gene (regions V1–V2, 454 platform) in fecal DNA from two representative mice colonized with *P. copri* revealed the ability of *Prevotella* to dominate the gut microbiota (*Figure 6B*). In comparison to fecal DNA from mice gavaged with media alone, *P. copri*-colonized mice had reduced Bacteroidales and Lachnospiraceae, similar to what was observed in this patient cohort (*Figure 1A*, *Figure 1—figure supplement 1D*). Consistent with a previous report of a *Prevotella* taxon exacerbating an inflammatory phenotype (*Elinav et al., 2011*), exposure of *P. copri*-colonized mice to 2% dextran sulfate sodium (DSS) in drinking water for 7 days resulted in more severe colitis as assessed by enhanced weight loss (*Figure 6C*), worse endoscopic score (*Figure 6D*), and increased epithelial damage on histological analysis (*Figure 6E,F*) when compared to littermate controls gavaged with media alone. Furthermore, in contrast to mice colonized with mouse commensal *Bacteroides thetaiotamicron* (*Figure 6—figure supplement 1A*), *P. copri* colonized mice similarly showed significantly decreased weight loss at day 7 following DSS exposure (*Figure 6—figure supplement 1B*). Analysis of the lamina propria CD4[+] T-cell response revealed an increase in IFNγ production following DSS induction, although no statistically significant differences were seen in IFNγ (Th1) or IL-17 production (Th17) following *P. copri* colonization (*Figure 6—figure supplement 1C*). Likewise, no differences in Foxp3[+] CD4[+] T-cells were observed. These data suggest that a *Prevotella*-defined microbiome may have the propensity to support inflammation in the context of a genetically susceptible host.

## Discussion

Multiple lines of investigation have revealed that RA is a multifactorial disease that occurs in sequential phases. Notably, there is a prolonged period of autoimmunity (i.e., presence of circulating auto-antibodies such as rheumatoid factor and anti-citrullinated peptide antibodies) in a pre-clinical state that lasts many years, during which time there is no clinical or histologic evidence of inflammatory arthritis (*Deane et al., 2010*). Before the onset of clinical disease, there is an increase in autoantibody titers and epitope spreading coupled with elevation in circulating pro-inflammatory cytokines. These findings have led to the 'second-event' hypothesis in RA, which proposes that an environmental factor triggers systemic joint inflammation in the context of pre-existent autoimmunity. Multiple mucosal sites and their residing microbial communities have been implicated, including the airways, the periodontal tissue and the intestinal lamina propria (*Mcinnes and Schett, 2011*; *Scher et al., 2012*).

Although a role for the gut microbiota has been clearly established in animal models of arthritis, it is not known if dysbiosis influences human RA. The human gut microbiota has been classified into unique enterotypes, one of which is defined by the predominance of *Prevotella* (*Arumugam et al., 2011*). In our cohort, we found the microbiota of many subjects to be defined by a single taxon—*P. copri*—which was associated with the majority of untreated, new-onset rheumatoid arthritis (NORA) patients. *P. copri* was also detected in a minority of healthy subjects in cohorts from the Human Microbiome Project (*Human Microbiome Project Consortium, 2012*), the European MetaHIT project (*Qin et al., 2010*), and our study. Surprisingly, the prevalence of *P. copri* in chronic rheumatoid arthritis (CRA) patients, all of whom had been treated and exhibited reduced disease activity, was similar to that observed in the healthy subjects. One hypothesis is that the *Prevotella*-defined microbiota fail to thrive

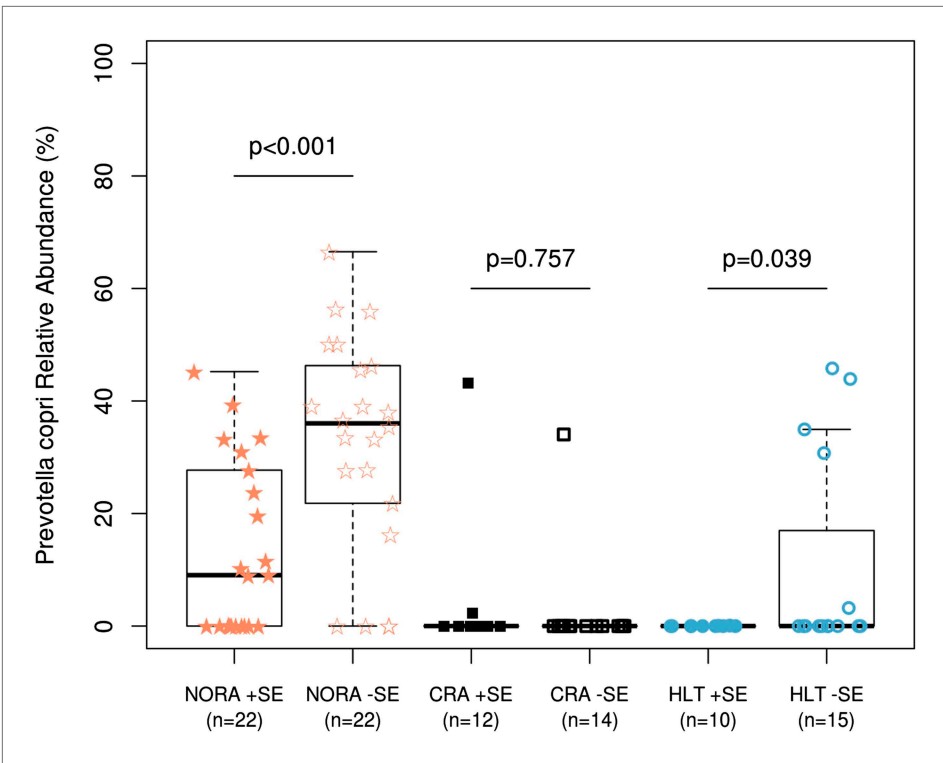

**Figure 5**. Relationship of host HLA genotype to abundance of *P. copri* (OTU4, OTU12, and OTU934 combined relative abundance). The HLA-class II genotype of all subjects was determined by sequence-based typing methodology ('Materials and methods'). Groups were subdivided by the presence or absence of shared-epitope RA risk alleles (+/− SE as indicated above) and correlated with relative abundance of intestinal *P. copri*. A statistically significant correlation is seen between *P. copri* abundance and the genetic risk for rheumatoid arthritis in NORA (red stars) and healthy (blue circles) subjects by Welch's two-tailed *t* test.
The following source data are available for figure 5:

**Source data 1**. Intermediate data and analysis tools for *Figure 5*.

when there is less inflammation, perhaps due to a lack of inflammation-derived terminal electron acceptors, as seen for *E. coli* in inflammatory bowel disease (*Winter et al., 2013*). Alternatively, the gut microbiota changes observed in newly diagnosed RA patients may be the consequence of a unique, NORA-specific systemic inflammatory response. While DAS28 scores were slightly lower in CRA and PsA patients (*Table 1*), the most remarkable difference was in levels of C-reactive protein (CRP). This raises the question of whether CRP itself may have microbial modulating properties. CRP is characteristically high in early and flaring RA, but not in other autoimmune diseases (e.g., systemic lupus erythematous, scleroderma, and PsA). A member of the pentraxin protein family, CRP was first identified in the plasma of patients with *Streptococcus pneumoniae* infection (*Tillett and Francis, 1930*). Further, the primary bacterial ligand for CRP is phosphocholine, a component of multiple bacterial cell-wall components, including lipopolysaccharides (LPS). CRP binding to bacterial phosphocholine activates the complement system and enhances phagocytosis by macrophages. Whether or not CRP itself represents a specific response to the presence of *P. copri* in NORA is an area of future investigation. Interestingly, *Prevotella*-dominated healthy omnivore individuals were recently reported to have increased basal levels of serum TMAO (trimethylamine N-oxide), a product of inflammation linked to atherogenesis, compared to *Bacteroides*-dominated healthy individuals (*Koeth et al., 2013*). While TMAO could be derived from increased consumption of meat (*Koeth et al., 2013*), *Prevotella* has been previously associated with a dearth of meat in the diet (*Wu et al., 2011*). Additional studies are needed to determine if prevalence of *P. copri* in the microbiota is associated with changes in specific metabolites.

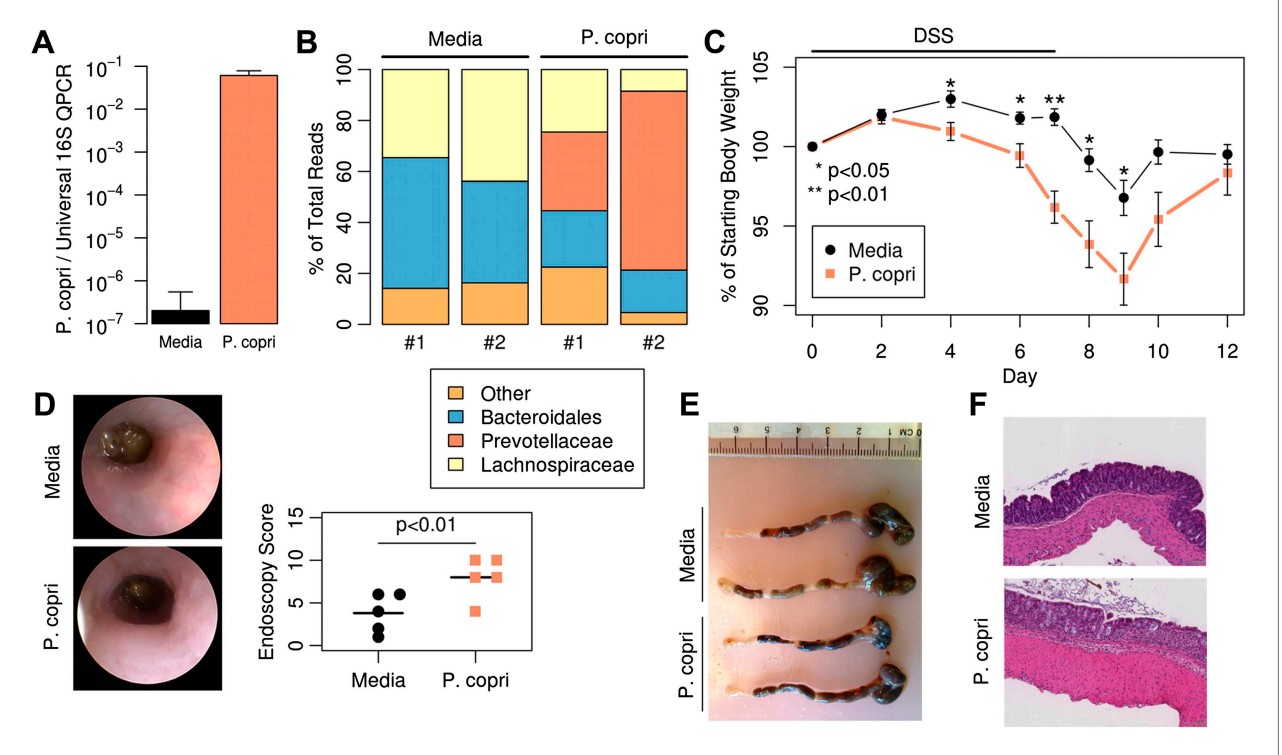

**Figure 6.** Colonization with *P. copri* dominates the colonic microbiome and exacerbates local inflammatory responses. (**A**) DNA was extracted from fecal pellets of media-gavaged mice and *P. copri*-gavaged mice 2 weeks after colonization and assayed by QPCR with *P. copri* specific primers compared to universal 16S. (**B**) Relative abundance of bacterial families in fecal DNA from media-gavaged and *P. copri*-colonized mice (shown in duplicate) by high-throughput 16S sequencing (regions V1–V2, 454 platform). (**C**) C57BL/6 mice colonized with *P. copri* (n = 15) or media alone (n = 13) controls were exposed to DSS for seven days and percent of starting body weight is shown. Composite data from three representative experiments are shown. (**D**) Representative colonoscopic images of mice colonized with *P. copri* or media gavage following DSS-induced colitis. Endoscopic colitis score for five individual animals is displayed. (**E** and **F**) Gross pathology (**E**) and histology (**F**) of colons from mice colonized with *P. copri* or media gavage following DSS-induced colitis.

The following source data and figure supplements are available for figure 6:

**Source data 1**. Intermediate data and analysis tools for **Figure 6**.

**Figure supplement 1**. *P. copri* colonization exacerbates chemically induced colitis.

Sequence alignment most closely linked NORA-associated *Prevotella* with the *P. copri* genome. Interestingly, large regions of the *P. copri* genome were scarcely covered in both our cohort and subjects of the HMP. As the reference strain of *P. copri* was isolated in Japan and all samples analyzed in our study were collected and sequenced in North America, these differences may reflect geographically-associated strain variability, consistent with a report ranking *P. copri* as the second-most variable member of the human gut microbiota between continents (*Schloissnig et al., 2013*). Notably, comparison of sequences in NORA samples with those of *P. copri*-dominated healthy individuals evaluated in the HMP allowed us to identify ORFs associated with the NORA phenotype. Two ORFs, both encoding components of an iron transporter, were specific for NORA-associated *P. copri*, while two ORFs were specific for HLT-associated *P. copri* and encode components of a *nuo* operon. Iron transporters are known to be virulence factors in other bacterial clades, while the ubiquinone oxidoreductase pathway encoded by the *nuo* operon may provide a fitness advantage in the context of a healthy microbiome by allowing use of metabolites available therein. While colonization with *P. copri* increases the pre-test probability of NORA from 1% to approximately 3.95% in western cohorts (by Bayes' theorem, see 'Materials and methods'), the presence of one of the aforementioned ORFs may markedly increase the pre-test probability of NORA status. The diagnostic application of these biomarkers needs to be confirmed in larger cohorts.

Analysis of enzymatic functions in the *Prevotella*-dominated metagenome reveals a significant decrease in purine metabolic pathways, including tetrahydrofolate (THF) biosynthesis. This may have therapeutic implications since methotrexate (MTX), a folate analogue and a dihydrofolate (DHF) reductase inhibitor, remains the anchor drug for the treatment of RA (*Singh et al., 2012*) and has inter-individual variability in terms of absorption and bioavailability. The THF biosynthetic pathway encoded by the gut metagenome, which includes a DHF reductase enzyme, may compete with host DHF reductase for MTX binding and metabolism. If so, an increase in DHF reductase-high microbiota in some RA subjects (i.e., *Bacteroides* overabundant) may help explain, at least partially, why only about half of RA patients respond adequately to oral MTX, ultimately requiring either parenteral administration or the addition of complementary immunosuppressants. *Prevotella*-high NORA subjects, with a dearth of DHF reductase in the gut, may respond better to oral MTX. Prospective human studies should help to clarify these observations.

RA is a multifactorial autoimmune disease in which certain alleles within the major histocompatibility complex (MHC) class II locus, specifically those belonging to DRB1 (i.e., shared epitope alleles), confer higher risk for disease. A recently published study with HLA-DR transgenic mice revealed that the gut microbiota was, at least partially, regulated by the HLA genes (*Gomez et al., 2012*). Arthritis-susceptible DRB1*04:01 transgenic mice had a markedly different intestinal microbiota when compared to arthritis-resistant DRB1*04:02 animals, and this was associated with altered mucosal immune function (i.e., increased gene transcripts for Th17-related cytokines) and increased intestinal permeability. Our results suggest that, similarly, SE risk-alleles in humans may have an impact on the composition of the gut microbiota. Intriguingly, patients in the NORA cohort showed a significant inverse correlation between *P. copri* relative abundance and presence of SE alleles (*Figure 5*). It is therefore possible that, as in mice, certain human gut microbial communities are determined by specific MHC alleles that favor the expansion of particular species. As in the case of cigarette smoking, this could also represent a gene-environment interaction that contributes to RA pathogenesis. It is conceivable that a certain threshold for *P. copri* abundance may be necessary to overcome the lack of genetic predisposition in RA subjects, while a lower abundance may be sufficient to trigger disease in those carrying risk-alleles. Validation in expanded cohorts and mechanistic studies are needed to better understand the significance of these findings.

Colonization of mice with *P. copri* recapitulated the differences in relative abundances of *Prevotella* and *Bacteroides* previously reported in humans, and confirmed the ability of *P. copri* to dominate the colonic commensal microbiota in the absence of apparent disease (*Faust et al., 2012*). This shift in abundances correlated with a metagenomic shift, which may support and/or perpetuate an inflammatory environment. For example, uniquely present superoxide reductase in *P. copri* may facilitate resistance to or allow the use of host-derived reactive oxygen species (ROS) generated during inflammation, perhaps as terminal electron acceptors for respiration (*Winter et al., 2013*). Similarly, the *P. copri* genome encodes phosphoadenosine phosphosulfate reductase (PAPS), an oxidoreductase absent in *Bacteroides* that participates in sulfur metabolism and leads to the production of thioredoxin. Intriguingly, thioredoxin has been widely implicated in the pathogenesis of RA and high levels of this redox protein have been found in both serum and synovial fluid of RA patients (*Maurice et al., 1999*).

Mice colonized with *P. copri* displayed increased inflammation in DSS-induced colitis. An appealing hypothesis from an evolutionary and ecological perspective is that the *P. copri*-defined microbiota thrives in a pro-inflammatory environment and may exacerbate inflammation for its own benefit. Another key feature of the *P. copri*-dominated microbiome is a community shift away from *Bacteroides*, Group XIV Clostridia, *Blautia*, and *Lachnospiraceae* clades, previously reported to be associated with an anti-inflammatory state and regulatory T-cell (Treg) production (*Atarashi et al., 2011*; *Round et al., 2011*). This could account, in part, for the observed differences in susceptibility to inflammation (*Tao et al., 2011*). Further characterization of changes in the host immune system associated with a *Prevotella*-dominated microbiota should provide deeper insight into whether expansion of *P. copri* contributes causally to the development of autoimmunity in early onset RA.

## Materials and methods

### Study participants

Consecutive patients from the New York University rheumatology clinics and offices were screened for the presence of RA. After informed consent was signed, each patient's medical history (according to chart review and interview/questionnaire), diet, and medications were determined. A screening

musculoskeletal examination and laboratory assessments were also performed or reviewed. All RA patients who met the study criteria were offered enrollment.

## Inclusion and exclusion criteria

The criteria for inclusion in the study required that patients meet the American College of Rheumatology/European League Against Rheumatism 2010 classification criteria for RA (*Aletaha et al., 2010*), including seropositivity for rheumatoid factor (RF) and/or anti–citrullinated protein antibodies (ACPAs) (assessed using an anti–cyclic citrullinated peptide ELISA; Euroimmun), and that all subjects be age 18 years or older. New-onset RA was defined as disease duration of a minimum of 6 weeks and up to 6 months since diagnosis, and absence of any treatment with disease-modifying anti-rheumatic drugs (DMARDs), biologic therapy or steroids (ever). Chronic RA was defined as any patient meeting the criteria for RA whose disease duration was a minimum of 6 months since diagnosis. Most subjects with chronic RA were receiving DMARDs (oral and/or biologic agents) and/or corticosteroids at the time of enrollment. Healthy controls were age-, sex-, and ethnicity-matched individuals with no personal history of inflammatory arthritis.

The exclusion criteria applied to all groups were as follows: recent (<3 months prior) use of any antibiotic therapy, current extreme diet (e.g., parenteral nutrition or macrobiotic diet), known inflammatory bowel disease, known history of malignancy, current consumption of probiotics, any gastrointestinal tract surgery leaving permanent residua (e.g., gastrectomy, bariatric surgery, colectomy), or significant liver, renal, or peptic ulcer disease. This study was approved by the Institutional Review Board of New York University School of Medicine.

## Sample collection and DNA extraction

Fecal samples were obtained within 24 hr of production. All samples were suspended in MoBio buffer-containing tubes. DNA was extracted using a combination of the MoBio Power Soil kit (Mo Bio Laboratories, Inc, Carlsbad, CA, USA) and a mechanical disruption (bead-beater) method based on a previously described protocol (*Ubeda et al., 2010*). Samples were stored at −80°C.

## V1–V2 16S rDNA region amplification and sequencing

For each sample, three replicate PCRs were performed to amplify the V1 and V2 regions as previously described (*Ubeda et al., 2010*). PCR products were sequenced on the 454 GS FLX Titanium platform (454 Life Sciences, Branford, CT, USA) to a depth of at least 2,600 reads per subject. Sequences have been deposited in the NCBI Sequence Read Archive under the accession number SRP023463.

## 16S sequence analysis

Sequence data were compiled and processed using MOTHUR (*Schloss et al., 2009*). Sequences were converted to standard FASTA format. Sequences shorter than 200 bp, containing undetermined bases or homopolymer stretches longer than 8 bp, with no exact match to the forward primer or a barcode, or that did not align with the appropriate 16S rRNA variable region were not included in the analysis. Using the 454 base quality scores, which range from 0–40 (0 being an ambiguous base), sequences were trimmed using a sliding-window technique, such that the minimum average quality score over a window of 50 bases never dropped below 30. Sequences were trimmed from the 3'-end until this criterion was met. Sequences were aligned to the 16S rRNA gene, using as template the SILVA reference alignment (*Pruesse et al., 2007*), and the Needleman-Wunsch algorithm with the default scoring options. Potentially chimeric sequences were removed using the ChimeraSlayer program (*Haas et al., 2011*). To minimize the effect of pyrosequencing errors in overestimating microbial diversity (*Huse et al., 2010*), rare abundance sequences that differ in one or two nucleotides from a high abundance sequence were merged to the high abundance sequence using the pre.cluster option in MOTHUR. Sequences were grouped into operational taxonomic units (OTUs) using the average neighbor algorithm. Sequences with distance-based similarity of 97% or greater were assigned to the same OTU. OTU-based microbial diversity was estimated by calculating the Shannon diversity index and Simpson Index using *mothur*. Phylogenetic classification was performed for each sequence using the Bayesian classifier algorithm described by Wang and colleagues with the bootstrap cutoff 60% (*Wang et al., 2007*).

## Statistical assessment of biomarkers using LEfSe

Briefly, LEfSe pairwise compares abundances of all biomarkers (e.g., bacterial clades) between all groups using the Kruskal-Wallis test, requiring all such tests to be statistically significant. Vectors resulting

from the comparison of abundances (e.g., *Prevotella* relative abundance) between groups are used as input to linear discriminant analysis (LDA), which produces an effect size (*Figure 1A*). In analyses performed here, the main utility of LEfSe over traditional statistical tests is that an effect size is produced in addition to a p or q value. This allows us to sort the results of multiple tests by the magnitude of the difference between groups, not only by q values, as the two are not necessarily correlated. In the case of hierarchically organized groups (e.g., bacterial clades, or KEGG pathways), this lack of correlation can arise from differences in the number of hypotheses considered at different levels in the hierarchy. For example, at the genus level, there may be 1,000 tests performed, requiring a high level of significance to pass multiple testing correction, whereas at the phylum level, only 10 tests may be performed, requiring a less stringent threshold for significance.

### Processing of Illumina reads

Paired-end reads 100 bp in length were trimmed from both ends to yield the largest contiguous segment where all per-base QVs were >= 25. Reads < 50 bp in length after this step were discarded. Quality-filtered reads were then aligned to the human reference genome (hg19) using bowtie2 in—very-sensitive-local mode, keeping only those reads that failed to align. Human-filtered reads were then sorted into complete pairs and singletons (whose mates were removed by filtering) for downstream analyses.

### Calculation of *P. copri DSM 18205* genome coverage

The *P. copri DSM 18205*-reference genome (assembly GCA_000157935.1) was first concatenated into a pseudo-contig in order of increasing contig number. Filtered Illumina reads from *P. copri* positive NORA and healthy (including HMP subjects, *Supplementary file 1A*) subjects were aligned to the reference using bowtie2 in—very-sensitive-local mode. Paired-end reads aligning to non-overlapping 1 kb windows across the length of the genome were counted and normalized to FPKM (fragments per kilobase per million reads). The interquartile range (25th to 75th percentile), mean, and median FPKM for each window was calculated and displayed as a boxplot with R.

### Generation of a *P. copri* pangenome catalog

Filtered paired-end reads from *P. copri* positive subjects were first assembled according to the HMP Whole-Metagenome Assembly SOP (*Pop, 2011*) using SOAPdenovo (*Luo et al., 2012*). Briefly, paired-end and singleton reads were used concurrently with the parameters -K 25 -R -M 3 -d 1. The resulting contigs >300 bp in length were then aligned to the *P. copri* reference genome with BLASTN at an e value cutoff of 1e-5. A stringent cutoff requiring at least one hit of 97% identity across 300 bp was used to infer that a contig originated from a strain of *P. copri* (*Figure 3—figure supplement 1D*). ORFs were then called on the resulting contigs using MetaGeneMark (*Zhu et al., 2010*). The resulting ORFs were then clustered using USEARCH at an identity threshold of 97% to yield a final set of *P. copri* genes (*Figure 3—figure supplement 1D*). Samples were excluded from further analyses if they had less than 7 million reads aligning to *P. copri* (*Figure 3—figure supplement 1C*). This resulted in a catalog of 20,387 putative *P. copri* ORFs with 9,274 +/− 1,640 (mean, SD) present in each subject. Further filtering of partially assembled (i.e., containing gaps, lacking stop codons), short (i.e., less than 300 bp), and low-coverage (i.e., present in fewer than five subjects) ORFs yielded a final set of 3,291 high-confidence *P. copri* ORFs.

### Presence or absence determination of *P. copri* pangenome ORFs

Filtered reads were aligned to the *P. copri* pangenome catalog using bowtie2 in–very-fast mode. ORFs were said to be present in a sample if at least 97% of their length, minus one read length (i.e., 100 bp) to account for edge alignment artifacts, was covered at an identity of 97% or greater (*Figure 3—figure supplement 1A*).

### Calculation of differential ORF presence in healthy and NORA

The presence or absence of ORFs in each sample was determined as above, and Fisher's exact test was used on 2 × 2 contingency tables for each ORF. Resulting p were adjusted for multiple hypothesis testing by converting to false discovery rate (FDR) q values using the Benjamini-Hochberg procedure. ORFs with q<0.25 were considered statistically significant. Effect size was calculated using the below equation.

$$Effect\,Size = \frac{Absent\,in\,NORA}{Total\,Absent} - \frac{Present\,in\,NORA}{Total\,Present}$$

## Application of Bayes' theorem to *P. copri* presence and NORA status

In western cohorts, such as the Human Microbiome Project and our own, the prevalence of *P. copri* is approximately 19%, that is P(*Prevotella*) = 0.19. The approximate incidence of RA is thought to be 1%, that is P(NORA) = 0.01. In our cohort, we found that 75% of new-onset RA (NORA) subjects had 5% or more *Prevotella* OTU4, which we determined to be *P. copri*, that is P(*Prevotella*|NORA) = 0.75. We therefore applied Bayes' theorem as given below.

$$P(NORA|Prevotella) = \frac{P(Prevotella|NORA)P(NORA)}{P(Prevotella)}$$

The solution to this equation gives a 3.95% probability of NORA status if *P. copri* is present in the gut, compared to a 1% probability of NORA (i.e., the incidence of RA) given no prior information.

## Genome assembly

Long reads were obtained for several high-*Prevotella* abundance subjects (028B, 030B, 061B, 089B) on the 454 GS FLX Titanium platform. These reads were assembled with Newbler v2.6 to obtain metagenomic assemblies (*Table 2*). The resulting contigs were subsequently filtered by alignment to the *P. copri DSM 18205* reference genome, keeping those with at least one hit of 97% across 300 bp, to obtain draft patient-derived *P. copri* genomes.

## Statistical significance of marker gene profiles between samplings

If each gene (boxes in *Figure 3B*, rows 61 boxes in length) is considered independently and can be in one of two states (i.e., present or absent), the probability of an exact match between any two individuals is $2^{-61}$, or $2^{-60}$ with one mismatch. Qualitatively, it can be seen that any intra- or inter-individual comparison is highly statistically significant. Further, if we concede that genes within an island are not truly independent, and there are six such islands which are considered identical with 1–2 mismatches allowed, the probability of such a match is $2^{-6}$, or 0.015625, less than a 0.05 threshold for significance.

## Quantification of metagenome function with HUMAnN and LEfSe

Filtered paired-end reads were aligned separately to all genomes in KEGG with USEARCH 6.0 (*Edgar, 2010*) using parameters—usearch_local—maxaccepts 2—maxrejects 8–evalue 0.1–id 0.80. The results from each read in a pair (and singletons) were combined and processed with HUMAnN 0.96 (*Abubucker et al., 2012*) with default parameters. Output tables containing per-sample abundance estimates of KEGG modules were then processed with LEfSe (*Segata et al., 2011*) using an alpha cutoff of 0.001 and an effect size cutoff of 2.0.

## Human leukocyte antigen (HLA) allele determination

Genomic DNA was isolated from the peripheral blood of RA patients and controls using QIAamp Blood Mini Kit (Qiagen GmbH, Halden, Germany) according to the manufacturer's instructions. HLA-DRB1 alleles were determined by Sequence-Based Typing (SBT) and by Single Specific Primer-Polymerase Chain Reaction (SSP-PCR) methodologies (Fred H Allen Laboratory of Immunogenetics, NY, USA; Weatherall Institute for Molecular Medicine, Oxford, UK) (*Supplementary file 1E*). Alleles considered to have the shared-epitope conferring higher risk for RA included: HLA-DRB1*01:01, 01:02, 04:01, 04:04, 04:05, 04:08, 10:01, 13:03, and 14:02, corresponding to $S_2$ and $S_{3P}$ RA risk classification (*du Montcel et al., 2005*). Subjects with at least one copy of these alleles have >1.95 times the relative risk of disease compared to the least at-risk genotype studied.

## Colonization of mice

C57BL/6 mice (Jackson Laboratories) were treated with ampicillin, neomycin, metronidazole (all 1 g/l) for 7 days prior to gavage. *P. copri* (CB7, DSMZ) or *B. thetaiotamicron* (gift from E Martens) was grown to log phase under anaerobic conditions in PYG liquid media (Anaerobe Systems, CA, USA) and $10^7$ CFU were used to inoculate mice. Feces were collected at 1 and 2 weeks post-gavage to confirm colonization. Fecal DNA was extracted with mechanical bead beating with 0.1 mm zirconia silica beads (Biospecs Inc.) in 2% SDS followed by phenol chloroform extraction. Confirmation of colonization was achieved with *P. copri* genome specific primers (F: CCGGACTCCTGCCCCTGCAA, R: GTTGCGCCA

GGCACTGCGAT); *Prevotella* 16S primers (F: CACRGTAAACGATGGATGCC, R: GGTCGGGTTGC AGACC), *B. thetaiotamicron* SusC (F: CACAACAGCCATAGCGTTCCA, R: ATCGCAAAAATAAGA TGGGCAAA) (Benjida et al JBC 2011), and Universal 16S Primers (F: ACTCCTACGGGAGGCAGCAGT, R: ATTACCGCGGCTGCTGGC). QPCR was performed with a Roche Lightcycler (Roche USA, South San Francisco, CA, USA) and the following cycling conditions: 9°C for 5 m, 40 cycles of 95°C for 10 s, and 60°C for 30 s, 72°C for 30 s. Genomic DNA from *P. copri* was used to generate a standard curve to quantitate ng of *P. copri* present per mg of total feces.

## DSS-induced colitis

Mice were given 2% dextran sulfate sodium (DSS) in drinking water *ad libitum* for 7 days. Body weight was evaluated every 1–2 days over 14 days. Colonic mucosal damage 0 to 3 cm proximal to the anal verge was evaluated by direct visualization using the Coloview (Karl Storz Veterinary Endoscopy, Tuttlingen, Germany). Endoscopic scoring was performed as previously described: assessment of colon thickening (0–3 points), fibrinization (0–3 points), granularity (0–3 points), morphology of the vascular pattern (0–3 points), and stool consistency (normal to unshaped; 0–3 points) (*Becker et al., 2006*).

## Cell isolation and intracellular staining

Lamina propria mononuclear cells were isolated from colonic tissue as previously described (*Diehl et al., 2013*). Cells were stimulated with phorbol myristate acetate and ionomycin with brefeldin for 4 hr and prepared as per manufacturer's instruction with Cytoperm/Cytofix (BD Biosciences) for intracellular cytokine evaluation of IL-17A (eBiosciences 17B7) and IFNγ (eBiosciences XMG1.2). For Foxp3 analysis, cells were fixed and permeabilized as per manufacturer's instructions (eBiosciences, Inc., San Diego, CA, USA) and stained intracellularly with anti-Foxp3 (FJK-16s).

## Source data

Source files for the figures and figure supplements have been uploaded to github (https://github.com/polyatail/scher_et_al_2013) and as *Figure 1—source data 1*, *Figure 1—source data 2*, *Figure 2—source data 1*, *Figure 2—source data 2*, *Figure 3—source data 1*, *Figure 3—source data 2*, *Figure 4—source data 1*, *Figure 5—source data 1*, and *Figure 6—source data 1*. Any future updates will be made available on GitHub.

## Acknowledgements

The authors would like to thank Pamela Rosenthal, Soumya Reddy, and Peter Izmirly for help in patient recruitment; Flo Pauli and Sarah Meadows (HudsonAlpha), Agnes Viale and Lauren Lipuma (MSKCC) for sequencing; Mukundan Attur (NYU) for help in sample preparation; Xiang Qin and Joseph Petrosino (Baylor Genome Center) for help with *Prevotella* sequencing; Eric Martens (U Michigan) for his gift of *Bacteroides* strains; Joe DeRisi (UCSF) for computational resources; and Gerard Honig, Gretchen Diehl and Elke Kurz (NYU) for early help with mouse and microbiology experiments.

## Additional information

### Funding

| Funder | Grant reference number | Author |
| --- | --- | --- |
| National Institutes of Health | 1RC2AR058986 | Eric G Pamer, Steven B Abramson, Dan R Littman |
| Howard Hughes Medical Institute | | Dan R Littman |
| National Institutes of Health | K23AR064318 | Jose U Scher |
| National Institutes of Health | R01AI042135 | Eric G Pamer |
| American Gastroenterological Association | | Randy S Longman |

| Funder | Grant reference number | Author |
|---|---|---|
| NSF Graduate Research Fellowship | 1144247 | Andrew Sczesnak |
| National Institutes of Health | R01HG005969 | Curtis Huttenhower |
| Danone Research | PLF-5972-GD | Curtis Huttenhower |

The funders had no role in study design, data collection and interpretation, or the decision to submit the work for publication.

## Author contributions

JUS, AS, RSL, Conception and design, Acquisition of data, Analysis and interpretation of data, Drafting or revising the article; NS, Analysis and interpretation of data, Drafting or revising the article; CU, Acquisition of data, Analysis and interpretation of data, Drafting or revising the article; CB, Acquisition of data, Analysis and interpretation of data; TR, VC, Acquisition of data, Drafting or revising the article; EGP, SBA, Conception and design, Analysis and interpretation of data; CH, DRL, Conception and design, Analysis and interpretation of data, Drafting or revising the article

## Ethics

Human subjects: Consecutive patients from New York University rheumatology clinics were offered enrollment in this study after informed consent was obtained. This study was approved by the Institutional Review Board of New York University School of Medicine (NYU IRB protocol H#09-0658).
Animal experimentation: All animal experiments were performed in accordance with approved protocols for the New York University Institutional Animal Care and Usage Committee (institutional number A3435-01, protocol #110602-03).

# Additional files

### Supplementary files

• Supplementary file 1. (**A**) Read statistics of sequenced samples included in and excluded from biomarker analyses. (**B**) Presence/absence, p-values and FDR statistics for differentially represented ORFs in the P. copri pangenome biomarker analysis, with annotations. (**C**) KOs present in *P. copri DSM 18205* but not in any *Bacteroides* accounting for at least 5% of the total microbiota in any subject of the Human Microbiome Project. (**D**) KOs present in all genomes available for *Bacteroides* accounting for at least 5% of the total microbiota in any subject of the Human Microbiome Project and not present in *P. copri DSM 18205*. (**E**) HLA-DRB1 alleles were determined for subjects in the cohort. Counts of RA risk alleles (shared epitope) are indicated as 0 for homozygotes not at risk, one for heterozygotes, and two for homozygotes at risk ('Materials and methods). Shared epitope alleles appear in bold.

### Major datasets

The following dataset was generated:

| Author(s) | Year | Dataset title | Dataset ID and/or URL | Database, license, and accessibility information |
|---|---|---|---|---|
| Scher, et al. | 2013 | Intestinal microbiota of patients with arthritis | PRJNA203810; http://www.ncbi.nlm.nih.gov/bioproject/?term=PRJNA203810 | Publicly available at the NCBI BioProject database (http://www.ncbi.nlm.nih.gov/bioproject). |

The following previously published dataset was used:

| Author(s) | Year | Dataset title | Dataset ID and/or URL | Database, license, and accessibility information |
|---|---|---|---|---|
| HMP Consortium | 2010 | NIH Human Microbiome Project | PRJNA43021; http://www.ncbi.nlm.nih.gov/bioproject/?term=PRJNA43021 | Publicly available at the NCBI BioProject database (http://www.ncbi.nlm.nih.gov/bioproject). |

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
