## [Decision Letter]

Thank you for sending your work entitled “*Prevotella copri* defines a metagenomic enterotype that correlates with enhanced susceptibility to arthritis” for consideration at *eLife*. Your article has been favorably evaluated by a Senior editor and 3 reviewers, one of whom is a member of our Board of Reviewing Editors.

The following individuals responsible for the peer review of your submission have agreed to reveal their identity: Diane Mathis, Reviewing editor.

The Reviewing editor and the other reviewers discussed their comments before reaching this decision, and the Reviewing editor has assembled the following comments to help you prepare a revised submission.

We have now received the three reviewers' comments on your manuscript “*Prevotella copri* defines a metagenomic enterotype that correlates with enhanced susceptibility to arthritis”. The reviewers all agreed that the manuscript is very interesting and potentially important. They also concurred that the data on mouse models are weak, primarily due to their marginal/questionable significance (see detailed comments below). The human data are rather more convincing, but would be improved by additional bioinformatic analyses (detailed below). Therefore, we invite you to submit a revised version that eliminates the mouse data and addresses the following issues:

1) The Introduction could be shortened, especially the discussion on Th17 cells, since the mechanistic studies showing that *Prevotella* specifically regulates Th17 cells are not conclusive. Also, human data on the role of IL-17 is really only robust for psoriasis, not RA or IBD.

2) The presentation is a bit unfocused. A large part of the paper involves genomic analyses that do not advance the main story. If the goal was to determine why *P. copri* increases in arthritis patients, there is very little in Figures 2, 3 and 4 or the main text to convincingly suggest a specific answer. These “side experiments” distract from the more central question as to the putative mechanism linking *P. copri* to disease.

3) The paper also suffers from scientific jargon. For example, use of the word “enterotype” in the title and text is quite different from its original application in the MetaHIT paper. We would recommend removing this word given recent concerns about the methodology used in the original studies (Wu et al., Science 2011; Koren et al., PLoS Comp Biol 2013), the inconsistent usage of this term in the field, the lack of any quantitative enterotype clustering in the current paper, and the focus of this study on *P. copri* instead of more general patterns in community structure. Other more minor offenders are “high-throughput 16S”, “dysbiotic”, and “clade diversity”.

4) The main text states that “NORA and healthy subjects form distinct clusters” based on Figure 1. This is clearly not the case, as the NORA subjects (stars) and healthy controls (circles) are distributed across the entire graph. A more accurate statement would be that samples cluster by *Prevotella* abundance irrespective of disease phenotype.

5) The prevalence of OTU4 in patients and controls is a key finding that is mentioned in the Abstract. This should be expanded upon and appropriate statistical tests need to be included.

6) The observation that “*P. copri* strains vary between individuals and retain their individuality over time” seems like an important point, especially in light of other recent findings (e.g., Faith et al. Science 2013). Figure 3 seems to qualitatively support this point, but no statistical testing is done. Are there quantitative and significant differences between individuals? What controls were done?

7) Figure 3: These groups don't hold up to multiple hypotheses so this panel and Table S3 should be removed.

8) The methotrexate discussion, albeit speculative, is really fascinating and a clearly novel aspect of this work. Are there any correlations with methotrexate usage and Bacteroides abundance? Any differences in efficacy? Additional bioinformatics here could be quite useful in designing follow-up studies.

9) The NORA samples are the only ones with high systemic inflammation, as indicated by CRP levels. So the correlation might be with that rather than with arthritis per se. This also raises the question as to whether the increased *P. copri* levels reflect cause or effect vis-à-vis the inflammation. These points should be discussed.

10) RA is an HLA-associated disease. Are the NORA and control individuals HLA-matched, which has been the norm in such studies? At least in mice, H-2 alleles impact the gut microbiota. If the cohorts aren't HLA-matched, could the authors do a correlation assessment with the individuals they have (assuming they have HLA-typed the cohort)?

11) The interpretation of data on the CIA model in this context is confounded by the fact that a bolus of mycobacterium (CFA) was injected together with collagen.

12) The data on the CIA model are weak. The differences are barely significant as shown, i.e., in Figure 5 and Figure S7b are AUC values statistically significant? Why are “data from 2 of 4 representative experiments” shown? What does it look like if all data are compiled?

13) The CIA studies require a control comparator, e.g., the *B. thetaiotamicron* used for the colitis experiments.

---

## [Author Response]

We appreciate the constructive comments and valuable points raised by the reviewers and the editor. We have now made changes and edits accordingly. Overall, we agree that ours represents an initial step to characterize a unique microbiome profile in human RA. Our study revealed a strong association of *P. copri* with RA, that, along with our metagenomic findings, should set the stage for future broader human studies (for replication and validation purposes) and, concomitantly, for mechanistic experiments aimed at gaining insights to address possible causation.

*1) The Introduction could be shortened, especially the discussion on Th17 cells, since the mechanistic studies showing that* Prevotella *specifically regulates Th17 cells are not conclusive. Also, human data on the role of IL-17 is really only robust for psoriasis, not RA or IBD*.

The Introduction has been shortened as requested, especially the paragraph detailing the role of Th17 cells in the intestinal lamina propria. However, we feel that some reference to T cells is necessary for the reader to understand the background leading up to the experiments that we report here. Specifically, the report that a single intestinal microbial commensal—SFB—can induce spontaneous arthritis in a germ-free mouse model through activation of lamina propria and peripheral Th17 cells served as the inspiration to look for a similar microbe and mechanism in humans.

*2) The presentation is a bit unfocused. A large part of the paper involves genomic analyses that do not advance the main story. If the goal was to determine why* P. copri *increases in arthritis patients, there is very little in*
Figures 2, 3 and 4
*or the main text to convincingly suggest a specific answer. These “side experiments” distract from the more central question as to the putative mechanism linking* P. copri *to disease*.

We agree that there is little data to convincingly suggest a specific answer as to why this association is observed. At this stage, we are seeking to discover appealing hypotheses that can be tested in future studies. To that end, Figure 2 demonstrates that the *Prevotella* in our cohort, known only by 16S sequence, is actually one specific taxon: *Prevotella copri*. Figure 3 allows for the possibility that unique genes encoded by these particular bacteria may influence the association, while Figure 4 provides data in support of the notion that *P. copri* thrives in an inflammatory environment, and may exacerbate inflammation. Experiments to uncover the mechanistic basis of this association will require considerably more work, and we would like to leave readers with a sense of what may be possible and what our best leads are for future investigation.

*3) The paper also suffers from scientific jargon. For example, use of the word “enterotype” in the title and text is quite different from its original application in the MetaHIT paper. We would recommend removing this word given recent concerns about the methodology used in the original studies (Wu et al., Science 2011; Koren et al., PLoS Comp Biol 2013), the inconsistent usage of this term in the field, the lack of any quantitative enterotype clustering in the current paper, and the focus of this study on* P. copri *instead of more general patterns in community structure. Other more minor offenders are “high-throughput 16S”, “dysbiotic”, and “clade diversity”*.

We agree with the reviewers that certain terms and semantics are important to better clarify our findings. In particular, we are aware that the word ‘enterotype’ has been questioned by recently published work. We have now removed references to enterotype from the title and text, as requested, and clarified instances in which we utilized terminology such as high-throughput 16S, dysbiosis, and clade diversity. In addition, we have sought to explain the utility of the various bioinformatics tools.

*4) The main text states that “NORA and healthy subjects form distinct clusters” based on*
Figure 1*. This is clearly not the case, as the NORA subjects (stars) and healthy controls (circles) are distributed across the entire graph. A more accurate statement would be that samples cluster by* Prevotella *abundance irrespective of disease phenotype*.

We have changed the sentence as requested to better characterize this finding.

*5) The prevalence of OTU4 in patients and controls is a key finding that is mentioned in the Abstract. This should be expanded upon and appropriate statistical tests need to be included*.

We have expanded upon this observation at the end of the first paragraph of the results section and performed chi-squared tests. Briefly, NORA v. HLT, CRA, and PsA are statistically significant (p<0.05), while pairwise comparisons between other groups are not significant.

*6) The observation that “*P. copri *strains vary between individuals and retain their individuality over time” seems like an important point, especially in light of other recent findings (e.g., Faith et al. Science 2013).*
Figure 3
*seems to qualitatively support this point, but no statistical testing is done. Are there quantitative and significant differences between individuals? What controls were done*?

We have updated the legend to this figure and our Methods to reflect statistical testing. Briefly, if each of 61 genes is considered independently and can be in one of two states (i.e., present or absent), the probability of an exact match between any two individuals is 2^-61^, or 2^-60^ with one mismatch. Qualitatively, it can be seen that any intra- or inter-individual comparison is highly statistically significant. Further, if we concede that genes within an island are not truly independent, and there are six such islands which are considered identical with 1–2 mismatches allowed, the probability of such a match is 2^-6^, or 0.015625, less than a 0.05 threshold for significance.

*7)*
Figure 3*: These groups don't hold up to multiple hypotheses so this panel and Table S3 should be removed*.

We assume the reviewers mean that the FDR-adjusted p-values (q-values) are not less than 0.05, the standard minimum required for statistical significance. In this instance, we feel that a higher threshold for significance is justified. FDR is intended to work in a different way than a standard p-value generated by, for example, a t-test. The threshold can be viewed as the percentage of biomarkers that are likely to be false positives. Given the number of hits returned in this analysis (i.e., 19), we expect only 4.75 to be false positives, with a great majority expected to be true. In exploratory analyses such as the one we conducted for this paper, a higher FDR threshold is often used—for example, the popular GSEA (Gene Set Enrichment Analysis) software uses an FDR cutoff of 0.25 by default (http://www.pnas.org/content/102/43/15545). Additionally, the four ORFs we chose for discussion are components of the same pathway, which appear adjacent to one another on the same metagenomic contigs. While it is difficult to devise a statistical test for such a situation, biological intuition suggests that these may be meaningful. We do concede that without validation of these biomarker ORFs our claims cannot be stated too strongly, and we have tried to phrase our conclusions as such.

*8) The methotrexate discussion, albeit speculative, is really fascinating and a clearly novel aspect of this work. Are there any correlations with methotrexate usage and Bacteroides abundance? Any differences in efficacy? Additional bioinformatics here could be quite useful in designing follow-up studies*.

The reviewers raise a very important point, namely that usage of methotrexate may be associated with both Bacteroides abundance and differences in treatment efficacy. The question of methotrexate efficacy, however, can only be addressed by prospective cohort design, as suggested by the reviewers. Similarly, and given the cross-sectional nature of our current study, the alteration of gut flora by the use of methotrexate cannot be answered by data accrued at this time. We agree with the reviewers that this is perhaps one of the most potentially relevant aspects of our work. We are currently engaged in prospective follow up studies to determine the effects of methotrexate in modulating gut microbiota.

*9) The NORA samples are the only ones with high systemic inflammation, as indicated by CRP levels. So the correlation might be with that rather than with arthritis* per se*. This also raises the question as to whether the increased* P. copri *levels reflect cause or effect vis-à-vis the inflammation. These points should be discussed*.

We thank the reviewers for raising these important questions. In fact, during the study design phase, we have discussed at length which disease would represent the most appropriate control group/s, specifically to address systemic inflammation as a possible modulator of the gut microbiome. In our study, NORA samples had (as expected) overall higher disease activity scores, reflecting untreated RA. In order to address “inflammation” as a confounder, we have included two reasonable positive control groups. CRA samples in our study have, by inclusion criteria, longer disease duration and have been under various treatment regimens at the time of enrollment. We have also enrolled recent-onset, mostly untreated PsA samples as our second control group. In both cases, as reflected in Table 1, disease activity scores were slightly lower than those found in the NORA group. Although we believe this comparison addresses the issue of systemic inflammation as modulator of gut microbiome, it is still possible that microbiota changes observed in newly diagnosed RA patients represent rather a consequence of a unique, NORA-specific systemic inflammatory response. A paragraph has now been included discussing this alternative possibility. A second, related issue that requires further investigation is the role of CRP in the modulation of microbiota. Importantly, while DAS28 scores were slightly lower in CRA and PsA patients, the most remarkable difference was found in levels of CRP. It is particularly intriguing to us whether CRP itself may have microbial modulating properties. CRP is synthesized by the liver in response to factors released by macrophages and adipocytes. It is a member of the pentraxin protein family and was first identified in the plasma of patients with *Streptococcus pneumoniae* infection and it was named according to its ability to precipitate the somatic C-fraction of the pneumococcal cell wall. Curiously, CRP was the first pattern recognition receptor (PRR) to be identified. The primary bacterial ligand for CRP is now recognized to be phosphocholine, a component of several bacterial cell wall structures. The physiological role of CRP consists in binding phosphocholine and the activation of the complement system leading to phagocytosis. Interestingly, and unlike many other autoimmune diseases (such as Systemic Lupus Erythematous (SLE), scleroderma, polymyositis, dermatomyositis and PsA), CRP is characteristically high in RA. Whether or not CRP itself represents a specific response to the presence of *P. copri* or other taxa is an area of future investigation. We have now added a paragraph addressing the reviewers’ comments.

*10) RA is an HLA-associated disease. Are the NORA and control individuals HLA-matched, which has been the norm in such studies? At least in mice, H-2 alleles impact the gut microbiota. If the cohorts aren't HLA-matched, could the authors do a correlation assessment with the individuals they have (assuming they have HLA-typed the cohort)*?

RA is considered a complex polygenic multifactorial autoimmune disease. Certain alleles within the HLA Class II locus confer higher risk for disease, in particular those belonging to DRB1 (i.e., shared epitope, or SE, alleles). However, genetic variance can only explain 20-30% of the cases. To address reviewers’ points, we have now included HLA-sequencing data. Indeed, consistent with recently published mouse data, the presence of SE risk-alleles seems to have an impact in the composition of gut microbiota. Our NORA cohort shows a significant inverse correlation between *P. copri* relative abundance and presence of shared epitope alleles. Intriguingly, a subgroup analysis of NORA patients according to presence/absence of SE alelles, revealed a significantly higher relative abundance of *P. copri* in those subjects lacking predisposing genes (P<0.001). It is possible therefore that, as in mice, microbiota abundance correlates with certain MHC alleles that favor an expansion of specific taxa. This could also represent a gene–environmental interaction for RA incidence, as reported for other factors such as smoking. Although we cannot prove causation, it is conceivable that a certain threshold of *P. copri* abundance may be necessary to overcome the lack of genetic predisposition in RA subjects, while a lower abundance may be sufficient to trigger disease in those carrying risk-alleles. Validation in expanded cohorts and mechanistic studies are needed to better understand the significance of these findings. A new figure and a paragraph expanding our findings are now included in the main text.

*11) The interpretation of data on the CIA model in this context is confounded by the fact that a bolus of mycobacterium (CFA) was injected together with collagen*.

The CIA results have been removed.

*12) The data on the CIA model are weak. The differences are barely significant as shown, i.e., in*
Figure 5
*and*
*Figure S7b*
*are AUC values statistically significant? Why are “data from 2 of 4 representative experiments” shown? What does it look like if all data are compiled*?

We agree; we have removed the CIA data.

*13) The CIA studies require a control comparator, e.g., the* B. thetaiotamicron *used for the colitis experiments*.

The CIA results have been removed.